# Hospitalization Costs of Lower Limb Ulcerations and Amputations in Patients with Diabetes in Romania

**DOI:** 10.3390/ijerph18052230

**Published:** 2021-02-24

**Authors:** Diana I. Sima, Cosmina I. Bondor, Ioan A. Vereşiu, Norina A. Gâvan, Cristina M. Borzan

**Affiliations:** 1Department of Diabetes and Metabolic Diseases, “Iuliu Haţieganu” University of Medicine and Pharmacy, 400006 Cluj Napoca, Romania; 2Department of Public Health, “Iuliu Haţieganu” University of Medicine and Pharmacy, 400083 Cluj Napoca, Romania; 3Department of Medical Informatics and Biostatistics, “Iuliu Haţieganu” University of Medicine and Pharmacy, 400349 Cluj Napoca, Romania; 4Podiatry Association, 400064 Cluj-Napoca, Romania

**Keywords:** diabetes mellitus, diabetic neuropathies, diabetic foot, amputation, economics, health care costs, tertiary prevention

## Abstract

In this retrospective case-control study conducted in Cluj-Napoca, Romania, we assessed the effect of ulcerations/amputations on hospitalization costs of patients with diabetes. Patients with (Group 1) or without (Group 2) ulcerations/amputations (case-control ratio 3:1) admitted to a single diabetes center between 2012–2017 were included. The effects of hospitalization days, age, duration of diabetes, body mass index and glycated hemoglobin (HbA1c) on total costs was explored using a multivariate linear regression analysis, enter model. Overall, 876 patients were included (Group 1: 682, 323 [47.4%] with amputations; Group 2: 194). Median (interquartile range) total expenses in Group 1 were 40% higher compared to Group 2 (€724 [504; 1186] vs €517 [362; 645], *p* < 0.001). Significant differences were observed between hospitalization costs (*p* < 0.001), cost of food (*p* < 0.001), medication (*p* = 0.044), drugs administered at the emergency room/intensive care unit (*p* < 0.001) and other expenses (*p* = 0.003). Hospitalization costs represented 80.5% of total expenses in Group 1 and 76.3% in Group 2. In multivariate analysis, hospitalization days influenced significantly the total costs in both groups (*p* < 0.001); in Group 2, the effect of HbA1c was also significant (*p* = 0.021). Diabetic foot ulcers and subsequent amputations most likely impose a significant economic burden on the Romanian public healthcare system.

## 1. Introduction

In the past decades, diabetes has evolved into a worldwide epidemic that poses a huge burden not only on an economic level, but also on the personal and social life of those affected by it [1]. Diabetes is one of the leading causes for lower limb amputations [2,3], and the risk is even higher in those with concurrent diabetic neuropathy [3,4]. Diabetes-related ulcerations are also associated with an increased risk of limb amputation and mortality, in addition to a lower quality of life [5]. Mortality rates amount to 40% in the first year after an ulceration/amputation and to over 80% in 5 years [6].

Treatment of patients with ulcers and amputations often poses considerable financial burden on health care systems [7,8]. Excess costs are mainly related to hospitalization, which is more frequent and often prolonged in patients with diabetes and ulcerations; in addition, use of resources involved in the clinical management of outpatients and post-amputation care further increase the costs [8,9]. In Germany, the mean cost incurred by the treatment of a patient with peripheral ulcers was €7537 in the year that the complication occurred. For patients undergoing amputation, annual mean costs reached €10,796, a more than 6-fold increase compared to patients with uncomplicated diabetes [7]. In England, costs of ulceration and amputation in diabetes in 2014–2015 were estimated at £837–962 million, representing 0.8% to 0.6% of the National Health Service’s expenditure. More than half of this amount (£501.48–£626.84 million) was spent on care of ulcerations in primary and community settings; 6.3% of hospital admissions in patients with documented diabetes included ulcer care or amputation [8]. Ignoring the costs of care may delay the implementation of effective strategies for prevention and management in both community and secondary care [8]. Total costs associated with ulcers and amputations include not only the direct expenses of hospitalization and medical care, but also the indirect ones generated by medical leave, supportive care, medical visits, use of total casts, delayed return to work and psychological support. 

According to data from the International Diabetes Foundation, more than 1,200,000 Romanian adults aged between 20 and 79 years have diabetes, and this number is expected to continue to rise [1]. Mean diabetes-related expenditure per person with diabetes in Romania is estimated at 2924.3 international dollars [1]; between 2008 and 2017, the mean (standard deviation [SD]) cost of diabetes incurred by the treatments of 667,384 (94,938) patients was €181,252 (74,278) [10]. Considering that since 2012, all costs with antidiabetic drugs and other expenditures (such as blood glucose tests, HbA1c assessment or insulin pumps/supplies) are reimbursed by the Romanian National Health Insurance House (CNAS), it is clear that the economic burden may represent a significant problem for the public health system. However, until now no data on excess costs associated with diabetic foot ulcers and amputations in Romania have been published. We hypothesize that ulcers and amputations significantly increase costs of healthcare in people with diabetes.

The primary objective of the study was to assess the costs of hospitalization of patients with diabetes with or without ulcerations, and to compare these costs according to demographic and clinical criteria. The secondary objective was to assess the association between ulcerations and the cost of hospitalization.

## 2. Materials and Methods

### 2.1. Study Design and Participants

This retrospective case-control study was conducted in Cluj-Napoca, Romania, in a single diabetes center that is part of a county hospital. Records of patients who were admitted between 2012 and 2017 were analyzed. Data from the patients’ records were collected from the institution’s electronic database by the main investigator. All patients who were admitted with a diagnosis of diabetes mellitus (“diabetes”) and of ulcerations (“ulcerations”) and/or amputations (“amputations”) in the lower limbs were included in Group 1. Controls were randomly selected among patients with diabetes, but without ulcerations or amputations, in a case-control ratio of 3:1 and included in Group 2. Exclusion criteria consisted of missing data on hospitalization duration and costs, or demographic data such as age and sex; patients who were admitted from emergency rooms (ER) for other acute complications (such as diabetic acidosis, myocardial infarction or stroke) were also excluded.

At admission, all patients were informed of the possibility of their anonymized data being used in a retrospective study and provided their written consent. The study protocol was revised and approved by the Ethics Committee of “Iuliu Haţieganu” University of Medicine and Pharmacy Cluj-Napoca, and by the Bioethics Committee Review Board (approval no. 341/01.10.2019).

### 2.2. Data Collection

Demographic data (age, sex, rural/urban residence), anthropometric data (weight, height, abdominal circumference), diabetes-related data (type, duration, glycated hemoglobin [HbA1c], treatment), details of diabetes complications (arteriopathy of the lower limbs, neuropathy, retinopathy, nephropathy, end-stage renal disease) and other co-morbidities were extracted from patients’ records, along with data regarding the duration and costs of hospitalization. Total hospitalization costs included expenses with accommodation (bed and bath), cost of food (three meals and a snack/day), medication (including also drugs used for the control of diabetes, which can only be prescribed by specialists, and drugs used in the emergency room), sanitary materials and paraclinical evaluations (blood tests, imaging). Costs related to any surgical procedure was not included. In Romania, all hospitalization costs incurred by insured patients are reimbursed through the National Health Insurance Fund, and patients receive a copy of the document with the costs at discharge (Table 1). For patients with multiple hospitalizations, costs for all episodes were aggregated.

### 2.3. Statistical Analysis

Absolute and relative frequencies were described for categorical variables. Normal distribution of the continuous variables was evaluated using the Kolmogorov–Smirnov test. Two means were compared using the Student’s-t test (for large samples and small samples with normal distribution) or the Mann–Whitney U test (for small samples with non-normal distribution); for comparing more than two means, the ANOVA t test (for large samples with equal variance and small samples with normal distribution) or the Kruskal-Wallis test (for samples with unequal variance and small samples with non-normal distribution) were used. Proportions were compared using the Chi-square test. To appreciate the relationship between two continuous variables, the Pearson coefficient of correlation was computed. A multivariate linear regression analysis, enter model, was performed with total costs as dependent variable and hospitalization days, age, duration of diabetes, BMI, and HbA1c as independent variables. To identify the factors influencing the duration of hospitalization and its cost (a major component of total cost), we divided the duration of hospitalization by the median number of hospitalization days in Group 1, and performed a multivariate logistic regression including nearly all studied parameters.

Level of statistical significance was set at 5%. All statistical analyses were performed using IBM SPSS Statistics 23 for Microsoft Windows.

## 3. Results

### 3.1. Patient Characteristics

Overall, 876 patients were included in the study, 682 in Group 1 and 194 in Group 2. No patients were excluded due to incomplete data. Significantly more patients in Group 1 were men (78.0%) compared to Group 2 (52.6%, *p* < 0.001); they were also significantly older (median age {range}: 64 {58,70} years vs. 62.5 {50,71} years in Group 1 and 2, respectively; *p* = 0.004) (Table 1). Median BMI was similar in both groups, and abdominal circumference was above the recommended average in European populations, with a median of 110 cm. Significantly more patients in Group 1 had type 2 diabetes, and diabetes duration was significantly longer. HbA1c levels were significantly lower in Group 1. Complications of diabetes were more frequent in Group 1—all patients in Group 1 have been previously diagnosed with diabetic neuropathy, compared to 2/3 of patients in Group 2 (*p* < 0.001); hypertension was similarly common in both groups (Table 2).

The majority of patients from both groups were under insulin treatment (70.6% and 88.1% in Groups 1 and 2, respectively), either alone, or combined with oral and/or injectable glucose lowering agents (GLA). 

Of the patients with ulcerations included in Group 1, 323 (47.4%) have suffered an amputation. Patients with ulcerations were also hospitalized for significantly longer compared to their counterparts without ulcerations (*p* < 0.001).

### 3.2. Cost Analysis

Median (interquartile range, IQR) total expenses incurred by a patient with diabetes and ulcerations was €724 (504; 1186), 40% higher compared to patients without ulcerations (€517 [362; 645], *p* < 0.001). Significant differences were observed between hospitalization costs (*p* < 0.001), cost of food (*p* < 0.001), medication (*p* = 0.044), drugs administered at the emergency room/intensive care unit (*p* < 0.001) and other expenses (*p* = 0.003), which were all higher in Group 1 (Figure 1). However, the costs of medication in the national program (*p* = 0.263), sanitary materials (*p* = 0.079) and costs with medical tests (*p* = 0.117) did not vary significantly between the two groups (Figure 1). The majority of patients with multiple hospitalizations were in Group 1: 188 (27.6%) patients vs. 2 (1%) in Group 2. The maximum number of hospitalizations recorded in one patient was 10.

The largest share of the expenses was represented by hospitalization costs in both groups: 80.5% in Group 1 and 76.3% in Group 2 (Figure 1). Hospitalization costs were influenced by the duration of hospitalization; as a consequence, there was a strong correlation between the number of hospitalization days and total costs in both groups (Group 1: r = 0.91, *p* < 0.001; Group 2: r = 0.97, *p* < 0.001). Total costs were correlated directly also with age (r = 0.20, *p* = 0.004) in Group 1. BMI, diabetes duration, and HbA1c did not significantly influence the total cost in any group in univariate analysis. This was in line with multivariate analysis results that showed a significant effect of hospitalization days (b = 79.71, 95%CI: 78.63–80.79, *p* < 0.001), but not of age, diabetes duration, BMI and HbA1c on the total costs in Group 1. In Group 2 however, in addition to hospitalization days (b = 85.43, 95%CI: 82.19–88.66, *p* < 0.001) the effect of HbA1c was also significant (b = −4.56, 95%CI: −8.42–−0.71, *p* = 0.021).

Sex, type of diabetes, treatments received, or the presence of hypertension, diabetic retinopathy, diabetic nephropathy and end stage renal disease did not influence total costs in either group. Peripheral arteriopathy was the only complication of diabetes associated with a higher total cost in Group 1 (Table 3). 

Hospitalization days, the parameter with the most important influence on costs, was correlated with diabetes duration (r = 0.13, *p* = 0.031) in Group 1 and with age (r = 0.23, *p* < 0.001) in Group 2. Higher HbA1c levels were independently associated with a higher number of hospitalization days (<7 days and /≥7) in both groups (Group 1: OR [95%CI] = 1.15 [1.01; 1.32], *p* = 0.034; Group 2: OR [95%CI] = 1.21 [1.04; 1.4], *p* = 0.015); while for diabetic nephropathy, the association was only significant in Group 1 (OR [95%CI] = 2.28 [1.12; 4.63], *p* = 0.023).

## 4. Discussion

The purpose of this study was to evaluate hospitalization costs incurred by patients with diabetes with or without ulcerations, the characteristics of the patients with ulcerations and the additional costs that come from the treatment of peripheral ulcers.

We found that total hospitalization costs were 40% higher in patients with diabetes who had ulcers and amputations, compared to those who did not. Even though the overall cost reported in this study is not as high as in most European countries, this may be because of undertreatment or lack of appropriate therapies. The total expenses with such patients are high for our median net income, which was the equivalent of €2049–2742 in the 2012–2017 period, compared to €15,469–16,924 in Europe [11]. Although direct comparisons cannot be made with the findings of other published studies [8,9,12], our results confirm the substantial economic burden related to foot ulcers and amputations. Petrakis et al. [12] reviewed costs associated with the treatment of diabetic foot ulcers and amputations reported in several European countries, USA, Canada, Australia and developing countries. Costs of managing diabetic foot ulcers differed significantly between countries, being up to an order of magnitude higher in developed countries compared to developing countries. Although study and population heterogeneity probably play a role in these differences, they are also likely to be the result of varying patterns of care and cost structures between countries [8].

In developed countries, inpatient costs of diabetic foot ulcer treatments, such as those assessed in our study, represented a major driver of diabetic foot-related costs [12,13]. Hospitalization costs are, in their turn, linked to hospitalization duration. As expected, the duration of hospitalization was significantly longer for patients with ulcers and amputations, contributing to the excess costs of diabetic foot ulcer management. This is not a singular finding, as significantly more hospitalization days were reported for patients with ulcers by several studies reviewed by Driver et al. [14].

Patients without ulcerations included in our study had significantly higher HBA1c levels compared to their counterparts with ulcerations. This finding is contrary to those reported in the literature, where higher HbA1c (an indicator of poor glycemic control) is usually reported as being a risk factor for diabetic foot ulcers [15]. One possible explanation for our results could be that many of the patients in the control group were admitted for treatment change due to inadequate glycemic control.

Peripheral arteriopathy is another well-known risk factor for diabetic foot ulcers; 49% of patients with ulcerations included in the EURODIALE study also had impaired lower limb perfusion, defined as an ankle-brachial pressure index of <0.9 and/or two absent foot pulses [16]. In our study, the percentage of patients with ulcerations/amputations and peripheral arteriopathy was somewhat lower, possibly reflecting underdiagnosis, since diagnostic tests for peripheral arteriopathy are not routinely performed. However, peripheral arteriopathy was the only complication associated with higher costs in the group of patients with ulcers. Considering that ischemia and deficient circulation lead to prolonged ulcer healing time and a higher risk of amputation, this finding is not surprising. After analyzing medical and pharmacy claims of 2253 patients with diabetes and ulcers, Stockl et al. [13] also found that patients who also had inadequate vascular status incurred higher total ulcer-related costs compared to their counterparts with adequate vascular status ($23,372 vs. $5218, respectively, *p* < 0.001). The most inexpensive way of reducing costs associated with lower extremity complications of diabetes is by preventing the underlying conditions that lead to them, like peripheral neuropathy, arteriopathy, poor glycemic control, low addressability towards physicians and poor hygienic control [17,18]. Several studies showed that strategies based on non-invasive testing can improve accuracy of diagnosing neuropathic complications of diabetes and, more importantly, decrease rates of amputations, at only slightly increased costs [19,20].

The average global prevalence of diabetic foot complications is 6.4% [21]. Foot ulcers and amputations are more common in low and middle-income countries [22], and it is considered that 25% of patients with diabetes in developing countries will develop at least one foot ulcer during their lifetime [5]. In Romania, Mota et al. [23] reported a prevalence of diabetes of 11.6%, based on the results of an epidemiological study conducted between 2012 and 2014; more than 1,200,000 Romanian adults in the 20–79 years age range are living with this disease [1,23]. The most recent data on the prevalence of diabetic foot complications comes from a study published in 2016, where 14.85% and 3.60% of 21,174 patients with diabetes reported a history of ulcers and amputations, respectively [24]. These numbers, combined with the additional costs of inpatient management generated by diabetic foot complications highlighted in our study, suggest that direct costs associated with management of foot ulcers could represent a considerable burden for the Romanian public health system. In addition, indirect costs that arise from medically related absenteeism, disability, social assistance and loss of work force, even though more difficult to assess, are also likely significant [12]. Furthermore, the negative societal and individual effects of diabetic foot complications, while not always quantifiable from an economic perspective, should not be ignored.

By having a thorough screening program, a multidisciplinary team formed of diabetologists, general practitioners, diabetes nurses, surgeons, radiology specialists and podiatrists adequately prepared and equipped to care for the patient with diabetes, lower extremity complications of diabetes could be greatly reduced. Ollendorf et al. [25] constructed a model to evaluate the effects of different interventions in preventing amputations in diabetic patients and found that by using educational intervention, a multidisciplinary team approach and therapeutic footwear 47–75% of amputations could be avoided. Management of the diabetic foot according to guideline-based care, patient education and a multidisciplinary team approach improves survival, reduces diabetic foot complications, is cost-effective, and even cost-saving compared with usual care, as demonstrated in theoretical models, as well as in real-life settings [26,27,28,29]. Thus, policy makers should place a higher emphasis on preventive and early interventions in the field of diabetic foot care [14].

Our study has some limitations. The analysis is based on data available from the electronic system, and the accuracy of it cannot be guaranteed. The center where study data were collected serves most of the central and northern part of the country, being one of the largest hospital units in the region. Patients with diabetes and diabetes-related complications are referred here from all hospitals in the surrounding counties, and some of these patients might be lost to follow-up over time, due to incapacity of travelling or being treated at a hospital closer to their home. In addition, as data were collected from a single center, their generalizability to the level of the entire country is limited. However, to our knowledge, this is the first study that explores cost-related aspects related to costs of diabetic foot ulcers and amputations in Romania; larger, multicenter studies would be needed to accurately reflect the economic burden of diabetic foot complications in our country.

## 5. Conclusions

By analyzing hospitalization costs of 876 patients with diabetes treated at a single Romanian center, we showed that patients with ulcers and amputations incurred significantly higher costs compared to their counterparts not exhibiting such complications. The largest driver of total hospitalizations costs was represented by the duration of hospitalization, which was significantly longer in patients with ulcerations. In this subset of patients, the presence of peripheral arteriopathy appears to also influence hospitalization costs significantly. Diabetic foot ulcers and subsequent amputations most likely impose a significant economic burden on the Romanian public healthcare system, emphasizing the need for health policies and strategies that target better prevention and care in the field of diabetic foot care.

## Figures and Tables

**Figure 1 ijerph-18-02230-f001:**
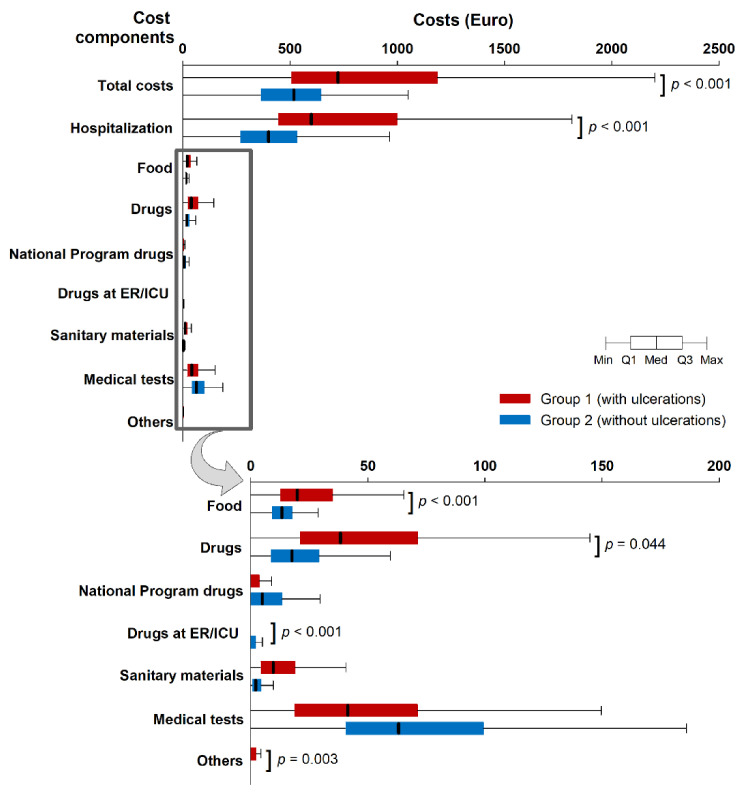
Comparison of hospitalization costs/patient in patients according to study group (Student-t test). Whiskers indicate non-outlier range.

**Table 1 ijerph-18-02230-t001:** Model of the patient expense statement completed at discharge.

Emergency Clinical County Hospital “Cluj Napoca”			
Patient Expense Statement			
Name, surname:			
Personal identification number:			
Address:			
Mail diagnosis at discharge:			
Hospitalization department:		File number:	
Period:	From:	To:	
Hospitalization days:			
Insurance status:			
1. Hospitalization costs (number of hospitalization days x cost/day):			
2. Food costs (food cost/one hospitalization day x number of days):			
3. Cost of drugs during hospitalization (art.3 from order 1100/2005):			
- Cost of drugs from the national health program:			
4. Cost of sanitary materials (art.3 from order 1100/2005):			
5. Cost of medical tests (art.3 from order 1100/2005):			
6. Other investigations (art.3 from order 1100/2005):			
- Cost of drugs in the Emergency Room:			
- Other medical services:			
**TOTAL COSTS:**			
Cost of drugs from the national health program:			
Written by:			
Date:			
Medical office:			

**Table 2 ijerph-18-02230-t002:** Baseline characteristics of study participants, according to the presence or absence of foot ulcerations.

Parameter	Group 1 (With Ulcerations) N = 682	Group 2 (Without Ulcerations) N = 194	*p*
**Male, *n* (%)**	532 (78.0)	102 (52.6)	<0.001
**Age, years ^1^**	64 (58–70)	62.5 (50–71)	0.004
**Type 2 diabetes, *n* (%)**	629 (92.2)	161 (83.0)	<0.001
**Diabetes duration, years ^1^**	14 (8–21)	8 (1–16)	<0.001
**BMI, kg/m^2^^1^**	28.63 (25.4–32.72)	28.41 (24.39–34.45)	0.068
**Abdominal circumference, cm ^1^**	110 (101–122)	110 (100.5–119)	0.091
**Hypertension, *n* (%)**	503 (73.8)	133 (70.0)	0.303
**HbA1c, %**	8.70 (7.50–10.25)	9.76 (8.00–12.00)	<0.001
**Peripheral arteriopathy, *n* (%)**	250 (36.7)	24 (12.4)	<0.001
**Diabetic neuropathy, *n* (%)**	682 (100)	133 (68.6)	<0.001
**Retinopathy, *n* (%)**	322 (47.2)	46 (23.7)	<0.001
**Nephropathy, *n* (%)**	108 (15.8)	10 (5.2)	<0.001
**End-stage renal disease, *n* (%)**	207 (30.4)	41 (21.1)	0.012
**Amputations, *n* (%)**	323 (47.4)	0 (0)	<0.001
**Hospitalization duration, days ^1^**	7 (5–12)	6 (4–8)	<0.001
**Treatment for diabetes**			<0.001
Diet, *n* (%) **^1^**	9 (1.3)	2 (1.0)	
Insulin, *n* (%) **^1^**	308 (45.2)	72 (37.3)	
GLA_O_, *n* (%) **^1^**	187 (24.7)	19 (9.8)	
GLA_O_+GLA_I_, *n* (%) **^1^**	5 (0.7)	2 (1.0)	
Insulin+GLA_O_, *n* (%) **^1^**	169 (24.8)	91 (47.2)	
Insulin+GLA_I_, *n* (%) **^1^**	2 (0.3)	1 (0.5)	
Insulin+GLA_O_+GLA_I_, *n* (%) **^1^**	2 (0.3)	6 (3.1)	

^1^ Median (25th–75th percentile). N, number of patients with available results; n (%), number (percentage) of patients in a given category; BMI, body mass index; HbA1c, glycated hemoglobin; GLA, glucose lowering agents; GLA_O_, oral GLA; GLA_I_, injectable GLA.

**Table 3 ijerph-18-02230-t003:** Cost analysis by study group.

Parameter	Category	Total Costs/Person, Euro (median [25th–75th percentile])		*p* **
		Group 1 (with Ulcerations) N = 682	Group 2 (without Ulcerations) N = 194	
**Sex**	Men	716.35 (503.79; 1141.15)	500.85 (319.21; 631.7)	<0.001
	Women	768.04 (507.96; 1307.76)	520.61 (370.69; 644.52)	<0.001
	*p* *	0.627	0.142	-
**Diabetes type**	Type 1	609.64 (495.3; 1086.53)	457.31 (319.21; 572.98)	0.001
	Type 2	739.11 (505.81; 1211.43)	520.61 (368.84; 652.58)	<0.001
	*p* *	0.322	0.268	-
**Hypertension**	Present	722.62 (503.79; 1154.82)	536.53 (372.1; 641.81)	<0.001
	Absent	734.82 (516.84; 1289.88)	465.82 (327.91; 660.63)	<0.001
	*p* *	0.317	0.458	-
**Peripheral arteriopathy**	Present	823.89 (546.35; 1376.01)	524.12 (366.2; 600.05)	<0.001
	Absent	664.79 (497.06; 1067.8)	517.01 (353.44; 664.32)	<0.001
	*p* *	0.003	0.590	-
**Diabetic neuropathy**	Present	723.51 (504.02; 1186.36)	501.52 (367.68; 630.33)	<0.001
	Absent	-	560.53 (326.11; 729.14)	-
	*p* *	-	0.312	-
**Retinopathy**	Present	668.52 (502.66; 1131.18)	471.52 (349.49; 579.57)	<0.001
	Absent	769 (505.15; 1241.86)	536.98 (368.84; 672.07)	<0.001
	*p* *	0.293	0.095	-
**Nephropathy**	Present	863.4 (555.84; 1615.88)	643.17 (454.53; 779.94)	0.031
	Absent	682.39 (502.66; 1131.18)	512.25 (357.5; 632.49)	<0.001
	*p* *	0.231	0.172	-
**End-stage renal disease**	Present	683.14 (503.2; 1256.8)	556.77 (394.14; 695.2)	0.001
	Absent	735.81 (505.37; 1136.98)	501.2 (361.57; 633.28)	<0.001
	*p* *	0.503	0.220	-
**Amputations**	Present	812.05 (529.05; 1323.55)	-	-
	Absent	648.59 (485.38; 1076.33)	-	-
	*p* *	0.308	-	-
**Treatment for diabetes**	Diet	705.4 (630.82; 873.79)	514.68 (195.7; 833.66)	0.582
	Insulin	755.11 (515.97; 1320.3)	539.99 (366.2; 712.13)	<0.001
	GLA_O_	681.65 (497.84; 1049.87)	556.77 (214.48; 642.29)	0.102
	GLA_O_+GLA_I_	809.17 (775.47; 988.1)	386.47 (257.98; 514.95)	0.190
	Insulin+GLA_O_	650.78 (498.45; 1100)	519.07 (394.15; 624.44)	<0.001
	Insulin+GLA_I_	731.22 (707.38; 755.06)	-	-
	Insulin+GLA_O_+GLA_I_	982.07 (656.38; 1307.76)	379.09 (348.32; 480.15)	0.071
	*p* *	0.685	0.449	

^1^ Median (25th–75th percentile); * within-group comparison; ** between-group comparison; N, number of patients with available results; GLA, glucose lowering agent; GLA_O_, oral GLA; GLA_I_, injectable GLA.

## Data Availability

The data that support the findings of this study are available from the corresponding author, upon reasonable request.

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
