# Peer review of "Hospitalization Costs of Lower Limb Ulcerations and Amputations in Patients with Diabetes in Romania"

_ijerph, 2021, doi:10.3390/ijerph18052230_

Round 1
Reviewer 1 Report
This study analyzed the effect of ulcerations/amputations on hospitalization costs of patients with diabetes. This paper addressed the interesting topic. In addition, it was written well. However, the contents of the conclusions are too short, and the contents of the main text are not enough. It is necessary to improve the quality of presentation by describing the results of this paper more specifically in the conclusions.Author Response
Reviewer #1
This study analyzed the effect of ulcerations/amputations on hospitalization costs of patients with diabetes. This paper addressed the interesting topic. In addition, it was written well.
- However, the contents of the conclusions are too short, and the contents of the main text are not enough. It is necessary to improve the quality of presentation by describing the results of this paper more specifically in the conclusions.
Response: We developed the conclusion section of the manuscript to make it more comprehensive:
“By analyzing hospitalization costs of 876 patients with diabetes treated at a single Romanian center, we showed that patients with ulcers and amputations incurred significantly higher costs compared to their counterparts not exhibiting such complications. The largest driver of total hospitalizations costs was represented by the duration of hospitalization, which was significantly longer in patients with ulcerations. In this subset of patients, the presence of peripheral arteriopathy appears to also influence hospitalization costs significantly. Diabetic foot ulcers and subsequent amputations most likely impose a significant economic burden on the Romanian public healthcare system, emphasizing the need for health policies and strategies that target better prevention and care in the field of diabetic foot care.”

Reviewer 2 Report
Congratulations to authors for studying this important issue raising awareness of economic implications of looking after patients with diabetes lower limb ulcerations/amputations.
Were duplicate patients (i.e. admitted on >1 occasion between 2012 and 2017) removed from analyses?
Do authors have data if amputations were toes / foot / below knee or above knee? Unilateral and/or bilateral?
Can authors clarify if patients were admitted without ulceration/amputation but subsequently required this during the same hospitalisation, how were they classified for the purposes of this study/analysis?
Suggest CONSORT diagram to show how many patients screened v.s. included v.s. excluded, which can help indicate if any selection bias.
Exclusion included those admitted for other acute complications but unclear what 'other acute complications' mean - does this mean the study only included patients who were admitted with the main reason for admission being ulcerations and/or amputations?
Importantly, this study highlights the very high prevalence of ulcerations/amputations (Group 1) in this population.
Interestingly, the HbA1c in group 1 was better than that in group 2 - can the authors think of possible explanations for this?
Is the prevalence of peripheral arteriopathy surprisingly low in groups 1 and 2? Of those with peripheral arteriopathy, do the authors have any data on how many of these patients had intervention?
Table 1 - suggest also include n(%) of those with ulcerations, and perhaps also n(%) of those with BOTH ulcerations and amputations? And any data on medications i.e. were any on antiplatelets, statins, insulin - some important medications were included in Table S2 but not in Table 1?
Table S2 - Nefropathy should be spelt Nephropathy
Figure 1 - Suggest perhaps adjusting scale so that easier to see costs for various categories (other than Hospitalization) - maybe Hospitalization could be on a separate graph on it's own, together with 'Total' - then footnote to explain different scale used for each graph.
Abstract has the year 2012-2107 but the year 2107 is obvious typo.
Author Response
Reviewer #2
Congratulations to authors for studying this important issue raising awareness of economic implications of looking after patients with diabetes lower limb ulcerations/amputations.
- Were duplicate patients (i.e. admitted on >1 occasion between 2012 and 2017) removed from analyses?
Response: For patients who had multiple admissions, we cumulated the costs accrued during all episodes of hospitalization to obtain the total hospitalization cost/patient. To clarify this, we made the following additions to the Methods section:
“In Romania, all hospitalization costs incurred by insured patients are reimbursed through the National Health Insurance Fund, and patients receive a copy of the document with the costs at discharge (Table 1). For patients with multiple hospitalizations, costs for all episodes were aggregated.”
- Do authors have data if amputations were toes / foot / below knee or above knee? Unilateral and/or bilateral?
Response: While we acknowledge that additional information on the level and localization of the amputations could have been valuable, we chose not to collect this data. This decision was taken because details regarding the amputations were not systematically included in all patient records.
- Can authors clarify if patients were admitted without ulceration/amputation but subsequently required this during the same hospitalisation, how were they classified for the purposes of this study/analysis?
Response: Group 1 (with ulcerations/amputations) included all patients who were admitted for this complication between 2012-2017, while Group 2 (without ulcerations/amputations) included only patients with diabetes but no ulcerations/amputations; these patients were selected from those who were admitted for periodic assessment of other chronic complications such as diabetic retinopathy or nephropathy. Therefore, a patient admitted without ulcerations but who subsequently required care for this complications or amputation during his hospitalization, would not have been included in either group.
- Suggest CONSORT diagram to show how many patients screened v.s. included v.s. excluded, which can help indicate if any selection bias.
Response: All patients admitted for the treatment of ulcerations between 2012-2017 and who also had cost-related and demographic data available were included in the study. Regarding the selection of the controls, this was done randomly, in a ratio of 3:1. Considering that no patient admitted for the treatment of ulcerations had incomplete data, and therefore virtually all screened patients were included in the analysis, we feel that a CONSORT diagram is not needed.
We acknowledge that the description of the method used to include patients in this study was not very clear, therefore we rephrased it to avoid confusion:
“Records of patients who were admitted between 2012 and 2017 were analyzed. Data from the patients’ records were collected from the institution’s electronic database by the main investigator. All patients who were admitted with a diagnosis of diabetes mellitus (“diabetes”) and of ulcerations (“ulcerations”) and/or amputations (“amputations”) in the lower limbs were included in Group 1. Controls were randomly selected among patients with diabetes, but without ulcerations or amputations, in a case-control ratio of 3:1 and included in Group 2.”
We also mentioned in the Results section that no patients were excluded due to incomplete data: “Overall, 876 patients were included in the study, 682 in Group 1 and 194 in Group 2. No patients were excluded due to incomplete data.”
- Exclusion included those admitted for other acute complications but unclear what 'other acute complications' mean - does this mean the study only included patients who were admitted with the main reason for admission being ulcerations and/or amputations?
Response: Indeed, Group 1 included only patients admitted for the treatment of ulcerations. Regarding the other acute complications mentioned in the text, we added a few examples in the Methods section:
“Exclusion criteria consisted of missing data on hospitalization duration and costs, or demographic data such as age and sex; patients who were admitted from emergency rooms (ER) for other acute complications (such as diabetic acidosis, myocardial infarction or stroke) were also excluded.”
- Importantly, this study highlights the very high prevalence of ulcerations/amputations (Group 1) in this population.
Response: Our study was not designed to assess the prevalence of ulcerations and/or amputations. While all patients admitted for the treatment of ulcerations between 2012-2017 were included in Group 1 (except for a few with missing data), not all patients without ulcerations/amputations admitted during the same period were included in Group 2. Controls (patients in Group 2) were selected randomly according to a 3:1 case-control ratio.
As mentioned above, we acknowledge that the description of the method used to include patients in this study was not very clear, therefore we rephrased it to avoid confusion:
“Records of patients who were admitted between 2012 and 2017 were analyzed. Data from the patients’ records were collected from the institution’s electronic database by the main investigator. All patients who were admitted with a diagnosis of diabetes mellitus (“diabetes”) and of ulcerations (“ulcerations”) and/or amputations (“amputations”) in the lower limbs were included in Group 1. Controls were randomly selected among patients with diabetes, but without ulcerations or amputations, in a case-control ratio of 3:1 and included in Group 2.”
- Interestingly, the HbA1c in group 1 was better than that in group 2 - can the authors think of possible explanations for this?
Response: Indeed, the finding of a seemingly better glycemic control in the group of patients with ulcerations was quite surprising. There are several possible explanations for this, including possibly a better compliance of patients who already developed complications or the effect of anemia in some patients. However, we believe that the most likely explanation for this difference is the fact that a large proportion of patients in the control group were admitted for treatment changes or adjustments, due to inadequate glycemic control. We included this possible explanation in the Discussion section as well: “Patients without ulcerations included in our study had significantly higher HBA1c levels compared to their counterparts with ulcerations. This finding is contrary to those reported in the literature, where higher HbA1c (an indicator of poor glycemic control) is usually reported as being a risk factor for diabetic foot ulcers [15]. One possible explanation for our results could be that many of the patients in the control group were admitted for treatment change due to inadequate glycemic control.”
- Is the prevalence of peripheral arteriopathy surprisingly low in groups 1 and 2? Of those with peripheral arteriopathy, do the authors have any data on how many of these patients had intervention?
Response: Our study was not designed to assess the prevalence of peripheral arteriopathy. However, we acknowledge that the true proportion of those with peripheral arteriopathy was probably higher, as some patients with less severe forms had not been diagnosed with PAD despite having this complication. We added a comment on this issue in the Discussion section:
“Peripheral arteriopathy is another well-known risk factor for diabetic foot ulcers; 49% of patients with ulcerations included in the EURODIALE study also had impaired lower limb perfusion, defined as an ankle-brachial pressure index of <0.9 and/or two absent foot pulses [16]. In our study, the percentage of patients with ulcerations/amputations and peripheral arteriopathy was somewhat lower, possibly reflecting underdiagnosis, since diagnostic tests for peripheral arteriopathy are not routinely performed. However, peripheral arteriopathy was the only complication associated with higher costs in the group of patients with ulcers.”
Peripheral vascular interventions in Romania are only performed in a few centers, meaning that only a few patients who would benefit from it have access to such procedures, unfortunately. While we did not collect this information, it is likely that the number of patients who had interventions is very low.
- Table 1 - suggest also include n(%) of those with ulcerations, and perhaps also n(%) of those with BOTH ulcerations and amputations? And any data on medications i.e. were any on antiplatelets, statins, insulin - some important medications were included in Table S2 but not in Table 1?
Response: Following the addition of supplementary material to the main manuscript, Table 1 has now become Table 2. All patients (100%) in Group 1 had ulcerations, as this was the main inclusion criteria (being admitted for the treatment of an ulcer), and no patients in Group 2 had ulcers, since only patients without ulcers were considered for inclusion as controls. The n(%) of those with both ulcers and amputations in Groups 1 and 2 is included in Table 2:
|
Amputations, n(%) |
323 (47.4) |
0 (0) |
We also added details regarding the treatment of patients to Table 2:
|
Treatment for diabetes |
|
|
<0.001 |
|
Diet, n(%) 1 |
9 (1.3) |
2 (1.0) |
|
|
Insulin, n(%) 1 |
308 (45.2) |
72 (37.3) |
|
|
GLAO, n(%) 1 |
187 (24.7) |
19 (9.8) |
|
|
GLAO+GLAI, n(%) 1 |
5 (0.7) |
2 (1.0) |
|
|
Insulin+GLAO, n(%) 1 |
169 (24.8) |
91 (47.2) |
|
|
Insulin+GLAI, n(%) 1 |
2 (0.3) |
1 (0.5) |
|
|
Insulin+GLAO+GLAI, n(%) 1 |
2 (0.3) |
6 (3.1) |
|
- Table S2 - Nefropathy should be spelt Nephropathy
Response: Following the addition of supplementary material to the main manuscript, Table S2 has now become Table 3. The misspelled word has been corrected.
|
Nephropathy |
Present |
863.4 (555.84; 1615.88) |
643.17 (454.53; 779.94) |
0.031 |
|
|
Absent |
682.39 (502.66; 1131.18) |
512.25 (357.5; 632.49) |
<0.001 |
|
|
p* |
0.231 |
0.172 |
|
- Figure 1 - Suggest perhaps adjusting scale so that easier to see costs for various categories (other than Hospitalization) - maybe Hospitalization could be on a separate graph on it's own, together with 'Total' - then footnote to explain different scale used for each graph.
Response: We redrew Figure 1 and presented data for lower costs separately, as suggested:
- Abstract has the year 2012-2107 but the year 2107 is obvious typo.
Response: The year was corrected: “…admitted to a single diabetes center between 2012-2017 were included.”

Reviewer 3 Report
- The manuscript could benefit from editing for grammar, missing words, and subject-verb agreement, etc. It is recommended that authors delete irrelevant "general" phrases and sentences, repeated and unneeded words. They should use short sentences. Also, some Introductory sentences are irrelevant or are not needed.
- Abstract: well-written.
- Title: It can be improved. Add “ulcerations/amputations” to be more specific.
- Figure 1 legend should be revised as to be more informative of the test/methodology used and main results presented. Specifically, please include the statistical test used, mean ± SD or SEM, etc. Also, add asterisk to Figure 1 to identify significance.
- Introduction: “In fact, diabetes is one of the leading causes for lower limb amputations.” Please replace the references [2] and [3] with updated ones.
- Introduction: In the second paragraph, I advise authors to elaborate a little bit on the reasons for high cost and how such complications pose high expenses on the hospitalization and management charges.
- Methods: well-presented.
- Results: what is meant by cost of food? Also, please add supplementary materials as main figures/tables in the text.
- Discussion section is well written. I suggest that authors focus on the main findings and avoid repeating results presentation in the discussion. Nevertheless, authors really did well in correlating their findings with what has been published in literature.
Author Response
Reviewer #3
- The manuscript could benefit from editing for grammar, missing words, and subject-verb agreement, etc. It is recommended that authors delete irrelevant "general" phrases and sentences, repeated and unneeded words. They should use short sentences. Also, some Introductory sentences are irrelevant or are not needed.
Response: We revised the document and tried to remove redundant parts and to clarify the potentially confusing ones. The Introduction section was revised as follows:
“In the past decades, diabetes has evolved into a worldwide epidemic that poses a huge burden not only on an economic level, but also on the personal and social life of those affected by it [1]. Diabetes is one of the leading causes for lower limb amputations [2,3], and the risk is even higher in those with concurrent diabetic neuropathy [3,4]. Diabetes-related ulcerations are also associated with an increased risk of limb amputation and mortality, in addition to a lower quality of life [5]. Mortality rates amount to 40% in the first year after an ulceration/amputation and to over 80% in 5 years [6].
Treatment of patients with ulcers and amputations often poses considerable financial burden on health care systems [7,8]. Excess costs are mainly related to hospitalization, which is more frequent and often prolonged in patients with diabetes and ulcerations; in addition, use of resources involved in the clinical management of outpatients and post-amputation care further increase the costs [8,9]. In Germany, the mean cost incurred by the treatment of a patient with peripheral ulcers was €7,537 in the year that the complication occurred. For patients undergoing amputation, annual mean costs reached €10,796, a more than 6-fold increase compared to patients with uncomplicated diabetes [7]. In England, costs of ulceration and amputation in diabetes in 2014–2015 were estimated at £837–962 million, representing 0.8% to 0.6% of the National Health Service’s expenditure. More than half of this amount (£501.48‒£626.84 million) was spent on care of ulcerations in primary and community settings; 6.3% of hospital admissions in patients with documented diabetes included ulcer care or amputation [8]. Ignoring the costs of care may delay the implementation of effective strategies for prevention and management in both community and secondary care [8]. Total costs associated with ulcers and amputations include not only the direct expenses of hospitalization and medical care, but also the indirect ones generated by medical leave, supportive care, medical visits, use of total casts, delayed return to work and psychological support.
According to data from the International Diabetes Foundation, more than 1,200,000 Romanian adults aged between 20 and 79 years have diabetes, and this number is expected to continue to rise [1]. Mean diabetes-related expenditure per person with diabetes in Romania is estimated at 2,924.3 international dollars [1]; between 2008 and 2017, the mean (standard deviation [SD]) cost of DM incurred by the treatments of 667,384 (94,938) patients with diabetes was €181,252 (74,278) [10]. Considering that since 2012, all costs with antidiabetic drugs and other expenditures (such as blood glucose tests, HbA1c assessment or insulin pumps/supplies) are reimbursed by the Romanian National Health Insurance House (CNAS), it is clear that the economic burden may represent a significant problem for the public health system. However, until now no data on excess costs associated with diabetic foot ulcers and amputations in Romania have been published. We hypothesize that ulcers and amputations significantly increase costs of healthcare in people with diabetes.
The primary objective of the study was to assess the costs of hospitalization of patients with diabetes with or without ulcerations, and to compare these costs according to demographic and clinical criteria. The secondary objective was to assess the association between ulcerations and the cost of hospitalization.”
- Abstract: well-written.
Response: Thank you. We corrected a typo in the Abstract: “Patients with (Group 1) or without (Group 2) ulcerations/amputations (case-control ratio 3:1) admitted to a single diabetes center between 2012-2017 were included.”
- Title: It can be improved. Add “ulcerations/amputations” to be more specific.
Response: We changed the title as suggested: “Hospitalization costs of lower limb ulcerations and amputations in patients with diabetes in Romania”
- Figure 1 legend should be revised as to be more informative of the test/methodology used and main results presented. Specifically, please include the statistical test used, mean ± SD or SEM, etc. Also, add asterisk to Figure 1 to identify significance.
Response: We added the statistical test used in the description of the figure. We also included additional details in the figure legend and presented p-values for the differences that were significant.
Figure 1. Comparison of hospitalization costs/patient in patients according to study group (Student-t test). Whiskers indicate non-outlier range.
- Introduction: “In fact, diabetes is one of the leading causes for lower limb amputations.” Please replace the references [2] and [3] with updated ones.
Response: We replaced the references with more recent ones:
Hussain, M.A.; Al-Omran, M.; Salata, K.; Sivaswamy, A.; Forbes, T.L.; Sattar, N.; Aljabri, B.; Kayssi, A.; Verma, S.; de Mestral, C. Population-based secular trends in lower-extremity amputation for diabetes and peripheral artery disease. CMAJ 2019, 191, E955-E961.
National Institute for Health and Care Excellence. Diabetic foot problems: prevention and management. 2015. Available online: https://www.nice.org.uk/guidance/ng19/resources/diabetic-foot-problems-prevention-and-management-pdf-1837279828933 (accessed on 19.02.2021).
- Introduction: In the second paragraph, I advise authors to elaborate a little bit on the reasons for high cost and how such complications pose high expenses on the hospitalization and management charges.
Response: We added additional details on the origin of excess costs, as requested:
“Treatment of patients with ulcers and amputations often poses considerable financial burden on health care systems [7,8]. Excess costs are mainly related to hospitalization, which is more frequent and often prolonged in patients with diabetes and ulcerations; in addition, use of resources involved in the clinical management of outpatients and post-amputation care further increase the costs [8,9].”
- Methods: well-presented.
Response: Thank you. We made some additions to the Methods section, to improve clarity:
“This retrospective case-control study was conducted in Cluj-Napoca, Romania, in a single diabetes center that is part of a county hospital. Records of patients who were admitted between 2012 and 2017 were analyzed. Data from the patients’ records were collected from the institution’s electronic database by the main investigator. All patients who were admitted with a diagnosis of diabetes mellitus (“diabetes”) and of ulcerations (“ulcerations”) and/or amputations (“amputations”) in the lower limbs were included in Group 1. Controls were randomly selected among patients with diabetes, but without ulcerations or amputations, in a case-control ratio of 3:1 and included in Group 2. Exclusion criteria consisted of missing data on hospitalization duration and costs, or demographic data such as age and sex; patients who were admitted from emergency rooms (ER) for other acute complications (such as diabetic acidosis, myocardial infarction or stroke) were also excluded. […] Demographic data (age, sex, rural/urban residence), anthropometric data (weight, height, abdominal circumference), diabetes-related data (type, duration, glycated hemoglobin [HbA1c], treatment), details of diabetes complications (arteriopathy of the lower limbs, neuropathy, retinopathy, nephropathy, end-stage renal disease) and other co-morbidities were extracted from patients’ records, along with data regarding the duration and costs of hospitalization. Total hospitalization costs included expenses with accommodation (bed and bath), cost of food (three meals and a snack/day), medication (including also drugs used for the control of diabetes, which can only be prescribed by specialists, and drugs used in the emergency room), sanitary materials and paraclinical evaluations (blood tests, imaging). Costs related to any surgical procedure was not included. In Romania, all hospitalization costs incurred by insured patients are reimbursed through the National Health Insurance Fund, and patients receive a copy of the document with the costs at discharge (Table 1). For patients with multiple hospitalizations, costs for all episodes were aggregated.”
- Results: what is meant by cost of food? Also, please add supplementary materials as main figures/tables in the text.
Response: Details regarding what is included in the cost of food are provided in the Methods section: “Total hospitalization costs included expenses with accommodation (bed and bath), cost of food (three meals and a snack/day), medication (including also drugs used for the control of diabetes, which can only be prescribed by specialists, and drugs used in the emergency room), sanitary materials and paraclinical evaluations (blood tests, imaging).”
Supplementary tables have been moved to the main document.
- Discussion section is well written. I suggest that authors focus on the main findings and avoid repeating results presentation in the discussion. Nevertheless, authors really did well in correlating their findings with what has been published in literature.
Response: We revised the Discussion section and removed parts that might have been perceived as repetitive. We also corrected a few typos:
“We found that total hospitalization costs were 40% higher in patients with diabetes who had ulcers and amputations, compared to those who did not. Even though the overall cost reported in this study is not as high as in most European countries, this may be because of undertreatment or lack of appropriate therapies. The total expenses with such patients are high for our median net income, which was the equivalent of €2,049‒2,742 in the 2012-2017 period, compared to €15,469‒16,924 in Europe [11]. Although direct comparisons cannot be made with the findings of other published studies [8,9,12], our results confirm the substantial economic burden related to foot ulcers and amputations. […] In addition, indirect costs that arise from medically related absenteeism, disability, social assisstance and loss of work force, even though more difficult to assess, are also likely significant [12]. Furthermore, the negative societal and individual effects of diabetic foot complications, while not always quantifiable from an economic perspective, should not be ignored.”
